# CAN ADVERSARIAL EXAMPLES BE PARSED TO REVEAL VICTIM MODEL INFORMATION?

## ABSTRACT

Numerous adversarial attack methods have been developed to generate imperceptible image perturbations that can cause erroneous predictions of state-of-the-art machine learning (ML) models, in particular, deep neural networks (DNNs). Despite intense research on adversarial attacks, little effort was made to uncover 'arcana' carried in adversarial attacks. In this work, we ask whether it is possible to infer data-agnostic *victim model* (VM) information (*i.e.*, characteristics of the ML model or DNN used to generate adversarial attacks) from data-specific adversarial instances. We call this 'model parsing of adversarial attacks' – a task to uncover 'arcana' in terms of the concealed VM information in attacks. We approach model parsing via supervised learning, which correctly assigns classes of VM's model attributes (in terms of architecture type, kernel size, activation function, and weight sparsity) to an attack instance generated from this VM. We collect a dataset of adversarial attacks across 7 attack types generated from 135 victim models (configured by 5 architecture types, 3 kernel size setups, 3 activation function types, and 3 weight sparsity ratios). We show that a simple model parsing network (MPN) is able to infer VM attributes from unseen adversarial attacks if their attack settings are consistent with the training setting (*i.e.*, in-distribution generalization assessment). We also provide extensive experiments to justify the feasibility of VM parsing from adversarial attacks, and the influence of training and evaluation factors in the parsing performance (*e.g.*, generalization challenge raised in out-of-distribution evaluation). We further demonstrate how the proposed MPN can be used to uncover the source VM attributes from transfer attacks, and shed light on a potential connection between model parsing and attack transferability.

## 1 INTRODUCTION

Adversarial attacks, in terms of tiny (imperceptible) input perturbations crafted to *fool* the decisions of machine learning (**ML**) models, have emerged as a primary security concern of ML in a wide range of vision applications (Szegedy et al., 2013; Goodfellow et al., 2014). Given the importance of the trustworthiness of ML, a vast amount of prior works have been devoted to answering the questions of ① *how to generate* adversarial attacks for adversarial robustness evaluation (Madry et al., 2017; Croce and Hein, 2020) and ② *how to defend* against these attacks for robustness enhancement (Zhang et al., 2019; Wong and Kolter, 2017). These two questions are also tightly connected: A solution to one would help address another.

In the plane of attack generation, a variety of attack methods were developed, ranging from gradient-based white-box attacks (Goodfellow et al., 2014; Carlini and Wagner, 2017; Croce and Hein, 2020) to query-based black-box attacks (Brendel et al., 2017; Chen et al., 2017; Liu et al., 2019). Understanding the attack generation process allows us to further understand attacks' characteristics and their specialties. For example, different from Deepfake images synthesized by generative models (Wang et al., 2020a; Asnani et al., 2021), adversarial attacks are typically determined by **(a)** a simple non-parametric and deterministic perturbation optimizer (*e.g.*, fast gradient sign method), **(b)** a specific input example (*e.g.*, an image), and **(c)** a specified, well-trained victim model (**VM**), *i.e.* an ML model against which attacks are generated. Here both (a) and (b) are interacted with and rely on VM for attack generation. The generated adversarial attacks in turn help the development of adversarial defenses. Examples include robust training (Madry et al., 2017; Zhang et al., 2019), adversarial detection (Metzen et al., 2017; Wang et al., 2020b), and adversarial purification (Yoon

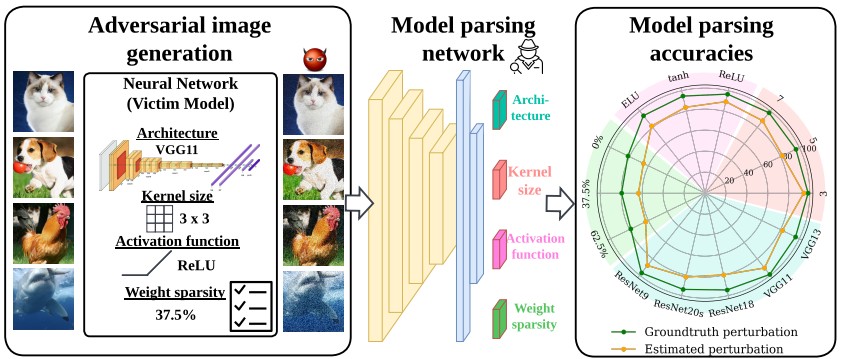

Figure 1: Schematic overview of victim model parsing from adversarial attacks. (Left) Attack generation based on victim model (with model attributes, architecture type, kernel size, activation function, and weight sparsity). (Middle) Proposed model parsing network (MPN) that assigns VM attribute class to input adversarial data. (Right) Highlighted results: The accuracy of model parsing from PGD attacks (Madry et al., 2017) against ResNet9 on CIFAR-10. Here the performance of MPN is reported under two different input data formats, the true adversarial perturbations and the estimated adversarial perturbations (proposed in Sec. 4).

et al., 2021; Nie et al., 2022), which exploit attack characteristics to recognize adversarial examples and produce anti-adversarial input perturbations.

In addition to ordinary attack generation and adversarial defense methods, some very recent works (Nicholson and Emanuele, 2023; Gong et al., 2022) started to understand and defend adversarial attacks in a new adversarial learning paradigm, termed reverse engineering of deception (**RED**). It aims to *infer* the adversary's information (*e.g.*, the attack objective and adversarial perturbations) from attack instances. Yet, nearly all the existing RED approaches focused on either estimation/attribution of adversarial perturbations (Gong et al., 2022; Goebel et al., 2021; Souri et al., 2021; Thaker et al., 2022) or recognition of attack classes/types (Nicholson and Emanuele, 2023; Wang et al., 2023; Maini et al., 2021; Zhou and Patel, 2022; Guo et al., 2023). None of the prior works investigated the feasibility of inferring *VM attributes* from adversarial attacks, given the fact that VM is the model foundation of attack generation. Thus, we ask (**Q**):

> *(Q) Are input-agnostic VM attributes invertible from input-specific adversarial attacks?*

We call problem (Q) **model parsing** of adversarial attacks; see **Fig. 1** for an illustration. This is also inspired by the model parsing problem defined for generative model (**GM**) (Asnani et al., 2021), which attempts to infer model hyperparameters of GM from synthesized photo-realistic images. However, adversarial attacks are data-specific input perturbations determined by hand-crafted optimizers rather than GM. And the 'model attributes' to be parsed from adversarial attacks are associated with the VM (victim model), which has a weaker correlation with attacked data compared to synthesized images by GM. The latter is easier to encode data-independent GM attribute information (Wang et al., 2020a; Asnani et al., 2021; Yu et al., 2019; Frank et al., 2020; Guarnera et al., 2020).

The significance of the proposed model parsing problem can also be demonstrated through a transfer attack example (**Fig. 2**). Suppose that adversarial attacks are generated from model *A* but used as transfer attacks against model *B*. If model parsing is possible, we will then be able to infer the true victim model source of these adversarial instances and shed light on the hidden model attributes.

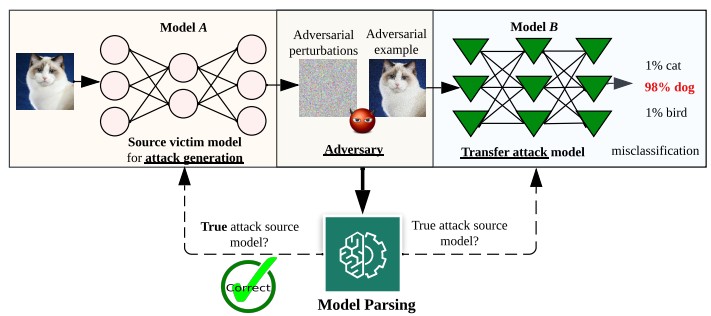

Figure 2: Motivating example on model parsing of transfer attack. A successful model parsing module can reveal the true source victim model.

**Contributions.** We summarize our contributions below.

- We formalize the problem of model parsing to infer VM attributes from adversarial attacks.

- We approach the model parsing problem of adversarial attacks as a supervised learning task and show that the learned model parsing network (MPN) could exhibit a surprising amount of generalization to recognize VM attributes from testing attack instances (see **Fig. 1** for highlighted results). We also peer into the influence of designing factors (including input data format, backbone network, and evaluation metric) in MPN's generalization.

- We make a comprehensive study on the feasibility of model parsing from adversarial attacks, including the in-distribution generalization on unseen adversarial images as well as the out-of-distribution generalization on unseen attack types and model architectures. We demonstrate how the model parsing approach can be used to uncover the source victim model attributes from transfer attacks (Fig. 2), and show connection between model parsing and attack transferability.

## 2 RELATED WORK

**Adversarial attacks and defenses.** Intensive research efforts have been made for the design of adversarial attacks and defenses. Adversarial attacks in the digital domain (Goodfellow et al., 2014; Carlini and Wagner, 2017) typically deceive DNNs by integrating carefully-crafted tiny perturbations into input data. Adversarial attacks in the physical domain (Eykholt et al., 2018; Li et al., 2019) are further developed to fool victim models under complex physical environmental conditions, which require stronger adversarial perturbations than digital attacks. The white-box attack typically leverages the local gradient information of VM to generate attacks (Goodfellow et al., 2014; Carlini and Wagner, 2017; Madry et al., 2017), while the black-box attack takes input-output queries of VM in attack generation; Examples include score-based attacks (Liu et al., 2019; Chen et al., 2017; Andriushchenko et al., 2020) and decision-based attacks (Brendel et al., 2017; Cheng et al., 2019; Chen and Gu, 2020). Given the vulnerability of ML models to adversarial attacks, methods to defend against these attacks are now a major focus in research. One line of research focuses on advancing model training methods to learn adversarially robust models (Madry et al., 2017; Zhang et al., 2019). Examples include min-max optimization-based adversarial training and its many variants, which can equip models with empirical robustness. To make models provably robust, certified training is also developed by integrating robustness certificate regularization into model training (Boopathy et al., 2021; Raghunathan et al., 2018) or leveraging randomized smoothing (Salman et al., 2020; 2019). In addition to training robust models, another line of research on adversarial defense is to detect adversarial attacks by exploring and exploiting the differences between adversarial data and benign data (Zhou and Patel, 2022; Grosse et al., 2017).

**Reverse engineering of deception (RED).** RED has emerged as a new adversarial learning to defend against adversarial attacks and infer the adversary's knowledge (such as its identifications, attack objectives, and attack perturbations). For example, a few recent works (Nicholson and Emanuele, 2023; Wang et al., 2023; Maini et al., 2021; Zhou and Patel, 2022; Guo et al., 2023) aim to reverse engineer the type of attack generation method and the used hyperparameters (*e.g.*, perturbation radius and step number). In addition, other works focus on estimating or attributing the adversarial perturbations used to construct adversarial images (Gong et al., 2022; Goebel et al., 2021; Souri et al., 2021; Thaker et al., 2022). This line of research also relates to adversarial purification (Srinivasan et al., 2021; Shi et al., 2021; Yoon et al., 2021; Nie et al., 2022), the technique to defend against adversarial attacks by characterizing and removing their harmful influence in model predictions. *However*, none of the prior works investigated if VM attributes are invertible from adversarial attacks. That is, the proposed model parsing problem remains open in adversarial learning. If feasible, the insight into model parsing could offer us an in-depth understanding of the encountered threat model and inspire new designs of adversarial defenses and robust models. Our work is also inspired by the model parsing problem in GM (generative model) (Asnani et al., 2021), aiming to infer GM attributes from their synthesized images. The rationale is that GM often encodes model fingerprints in synthesized images so that these fingerprints can be leveraged for DeepFake detection and model parsing (Wang et al., 2020a; Asnani et al., 2021; Yu et al., 2019; Frank et al., 2020; Guarnera et al., 2020). Lastly, we stress that RED is different from the work (Oh et al., 2019; Wang and Gong, 2018) on reverse engineering of (black-box) model hyperparameters, which estimates model attributes from the model's prediction logits. However, in this work VM is unknown for model parsing of adversarial attacks, and the only information we have is the dataset of attack instances.

## 3 PRELIMINARIES AND PROBLEM SETUPS

**Preliminaries: Adversarial attacks and victim models.** We first introduce different kinds of adversarial attacks and exhibit their dependence on **VM** (**victim model**), *i.e.*, the ML model from which attacks are generated. Throughout the paper, we will focus on $\ell_p$ attacks, where the adversary aims to generate imperceptible input perturbations to fool an image classifier (Goodfellow et al., 2014). Let $\mathbf{x}$ and $\boldsymbol{\theta}$ denote a benign image and the parameters of VM. The **adversarial attack** (*a.k.a*, adversarial example) is defined via the linear perturbation model $\mathbf{x}' = \mathbf{x} + \boldsymbol{\delta}$, where $\boldsymbol{\delta} = \mathcal{A}(\mathbf{x}, \boldsymbol{\theta}, \epsilon)$ denotes **adversarial perturbations**, and $\mathcal{A}$ refers to an attack generation method relying on $\mathbf{x}$, $\boldsymbol{\theta}$, and the attack strength $\epsilon$ (*i.e.*, the perturbation radius of $\ell_p$ attacks).

We focus on 7 attack methods given their different dependencies on the victim model ($\boldsymbol{\theta}$), including input gradient-based white-box attacks with full access to $\boldsymbol{\theta}$ (FGSM (Goodfellow et al., 2014), PGD (Madry et al., 2017), CW (Carlini and Wagner, 2017), and AutoAttack or AA (Croce and Hein, 2020)) as well as query-based black-box attacks (ZO-signSGD (Liu et al., 2019), NES (Ilyas et al., 2018), and SquareAttack or Square (Andriushchenko et al., 2020)).

Table 1: Summary of adversarial attack types focused in this work. Here GD refers to gradient descent, and WB and BB refer to white-box and black-box dependence on the victim model, respectively.

| Attacks | Generation method | Loss | $\ell_p$ norm | Strength $\epsilon$ | Dependence on $\theta$ |
|---|---|---|---|---|---|
| FGSM | one-step GD | CE | $\ell_\infty$ | {4, 8, 12, 16}/255 | WB, gradient-based |
| PGD | multi-step GD | CE | $\ell_\infty$ $\ell_2$ | {4, 8, 12, 16}/255 0.25, 0.5, 0.75, 1 | WB, gradient-based |
| CW | multi-step GD | CW | $\ell_2$ | soft regularization $c \in \{0.1, 1, 10\}$ | WB, gradient-based |
| AutoAttack or AA | attack ensemble | CE / DLR | $\ell_\infty$ $\ell_2$ | {4, 8, 12, 16}/255 0.25, 0.5, 0.75, 1 | WB, gradient-based + BB, query-based |
| SquareAttack or Square | random search | CE | $\ell_\infty$ $\ell_2$ | {4, 8, 12, 16}/255 0.25, 0.5, 0.75, 1 | BB, query-based |
| NES | ZOO | CE | $\ell_\infty$ | {4, 8, 12, 16}/255 | BB, query-based |
| ZO-signSGD | ZOO | CE | $\ell_\infty$ | {4, 8, 12, 16}/255 | BB, query-based |

✦ FGSM (fast gradient sign method) (Goodfellow et al., 2014): This attack method is given by $\boldsymbol{\delta} = \mathbf{x} - \epsilon \times \text{sign}(\nabla_{\mathbf{x}} \ell_{\text{atk}}(\mathbf{x}; \boldsymbol{\theta}))$, where $\text{sign}(\cdot)$ is the entry-wise sign operation, and $\nabla_{\mathbf{x}} \ell_{\text{atk}}$ is the input gradient of an attack loss $\ell_{\text{atk}}(\mathbf{x}; \boldsymbol{\theta})$ evaluated at $\mathbf{x}$ under $\boldsymbol{\theta}$.

✦ PGD (projected gradient descent) (Madry et al., 2017): This extends FGSM via an iterative algorithm. Formally, the $K$-step *PGD $\ell_\infty$ attack* is given by $\boldsymbol{\delta} = \boldsymbol{\delta}_K$, where $\boldsymbol{\delta}_k = \mathcal{P}_{\|\boldsymbol{\delta}\|_\infty \leq \epsilon}(\boldsymbol{\delta}_{k-1} - \alpha \times \text{sign}(\nabla_{\mathbf{x}} \ell_{\text{atk}}(\mathbf{x}; \boldsymbol{\theta})))$ for $k = 1, \ldots, K$, $\mathcal{P}_{\|\boldsymbol{\delta}\|_\infty \leq \epsilon}$ is the projection operation onto the $\ell_\infty$-norm constraint $\|\boldsymbol{\delta}\|_\infty \leq \epsilon$, and $\alpha$ is the attack step size. By replacing the $\ell_\infty$ norm with the $\ell_2$ norm, we similarly obtain the *PGD $\ell_2$ attack* (Madry et al., 2017).

✦ CW (Carlini-Wager) attack (Carlini and Wagner, 2017): Similar to PGD, CW calls iterative optimization for attack generation. Yet, CW formulates attack generation as an $\ell_p$-norm regularized optimization problem, with the regularization parameter $c = 1$ by default. For example, the choice of $c = 1$ in CW $\ell_2$ attack could lead to a variety of perturbation strengths with the average value around $\epsilon = 0.33$ on the CIFAR-10 dataset. Moreover, CW adopts a hinge loss to ensure the misclassification margin. We will focus on CW $\ell_2$ attack.

✦ AutoAttack (or AA) (Croce and Hein, 2020): This is an ensemble attack that uses AutoPGD, an adaptive version of PGD, as the primary means of attack. The loss of AutoPGD is given by the difference of logits ratio (DLR) rather than CE or CW loss.

✦ ZO-signSGD (Liu et al., 2019) and NES (Ilyas et al., 2018): They are zeroth-order optimization (ZOO)-based black-box attacks. Different from the aforementioned white-box gradient-based attacks, the only interaction mode of black-box attacks with the victim model ($\boldsymbol{\theta}$) is submitting inputs and receiving the corresponding predicted outputs. ZOO then uses these input-output queries to estimate input gradients and generate adversarial perturbations. Yet, ZO-signSGD and NES call different gradient estimators in ZOO (Liu et al., 2020).

✦ SquareAttack (or Square) (Andriushchenko et al., 2020): This attack is built upon random search and thus does not rely on input gradient.

Optimization methods, attack losses, $\ell_p$ norms, and dependencies on $\boldsymbol{\theta}$, are summarized in **Table 1**.

**Model parsing of adversarial attacks.** It is clear that adversarial attacks contain the information of VM ($\boldsymbol{\theta}$), although the degree of their dependence varies. Inspired by the above, one may wonder if the *attributes* of $\boldsymbol{\theta}$ can be *inferred* from these attack instances, *i.e.*, adversarial perturbations, or perturbed images. The model attributes of our interest include model architecture types as well as

finer-level knowledge, *e.g.*, activation function type. We call the resulting problem **model parsing of adversarial attacks**, as described below.

> **(Problem statement)** Is it possible to infer victim model information from adversarial attacks? And what factors will influence such model parsing ability?

To our best knowledge, the feasibility of model parsing from adversarial attack instances is an open question. Its challenges stay in two dimensions. **First**, through the **model lens**, VM (victim model) is indirectly coupled with adversarial attacks, *e.g.*, via local gradient information or model queries. Thus, it remains elusive what VM information is fingerprinted in adversarial attacks and impacts the feasibility of model parsing. **Second**, through the **attack lens**, the diversity of adversarial attacks (Table 1) makes a once-for-all model parsing solution extremely difficult. Spurred by the above, we will take the first solid step to investigate the feasibility of model parsing from adversarial attacks and study what factors may influence the model parsing performance. The insight into model parsing could offer us an in-depth understanding of the encountered threat model and inspire new designs of adversarial defenses and robust models.

**Model attributes and setup.** We specify VMs of adversarial attacks as convolutional neural network (CNN)-based image classifiers. More concretely, we consider 5 CNN architecture types (`AT`s): ResNet9, ResNet18, ResNet20, VGG11, and VGG13. Given an `AT`, CNN models are then configured by different choices of kernel size (`KS`), activation function (`AF`), and weight sparsity (`WS`). Thus, a valued quadruple (`AT`, `KS`, `AF`, `WS`) yields a specific VM ($\boldsymbol{\theta}$). Although more attributes could be considered, the rationale behind our choices is given below. We focus on `KS`

Table 2: Summary of model attributes of interest. Each attribute value corresponds to an attribute class in model parsing.

| Model attributes | Code | Classes per attribute |
|---|---|---|
| Architecture type | AT | ResNet9, ResNet18 ResNet20, VGG11, VGG13 |
| Kernel size | KS | 3, 5, 7 |
| Activation function | AF | ReLU, tanh, ELU |
| Weight sparsity | WS | 0%, 37.5%, 62.5% |

and `AF` since they are the two fundamental building components of CNNs. Besides, we choose `WS` as another model attribute since it relates to sparse models achieved by pruning (*i.e.*, removing redundant model weights) (Han et al., 2015; Frankle and Carbin, 2018). We defer sanity checks of all-dimension model attributes for future studies.

**Table 2** summarizes the focused model attributes and their values to specify VM instances. Given a VM specification, we generate adversarial attacks following attack methods in Table 1. Unless specified otherwise, our empirical studies will be mainly given on `CIFAR-10` but experiments on other datasets will also be provided in Sec. 5.

## 4 METHODS

In this section, we approach the model parsing problem as a standard supervised learning task applied over the dataset of adversarial attacks. We will show that the learned model parsing network could exhibit a surprising amount of generalization on test-time adversarial data. We will also show data-model factors that may influence such generalization.

**Model parsing network and training.** We approach the model parsing problem as a supervised attribute recognition task. That is, we develop a parametric model, termed model parsing network (**MPN**), which takes adversarial attacks as input and predicts the model attribute values (*i.e.*, 'classes' in Table 2). Despite the simplicity of supervised learning, the construction of MPN is non-trivial when designing data format, backbone network, and evaluation metrics.

We first create a model parsing dataset by collecting adversarial attack instances against victim models. Since adversarial attacks are proposed for evading model predictions in the post-training stage, we choose the test set of an ordinary image dataset (*e.g.*, `CIFAR-10`) to generate adversarial data, where an 80/20 training/test split is used for MPN training and evaluation. Following notations in Sec. 3, the training set of MPN is denoted by $\mathcal{D}_{\mathrm{tr}} = \{(\mathbf{z}(\mathcal{A}, \mathbf{x}, \boldsymbol{\theta}), y(\boldsymbol{\theta})) \mid \mathbf{x} \in \mathcal{I}_{\mathrm{tr}}, \boldsymbol{\theta} \in \Theta\}$, where $\mathbf{z}$ signifies an attack data feature (*e.g.*, adversarial perturbations $\boldsymbol{\delta}$ or adversarial example $\mathbf{x}'$) that relies on the attack method $\mathcal{A}$, the original image sample $\mathbf{x}$, and the VM $\boldsymbol{\theta}$, and $y(\boldsymbol{\theta})$ denotes the true model attribute label associated with $\boldsymbol{\theta}$. To differentiate with the testing data of MPN, we denote by $\mathcal{I}_{\mathrm{tr}}$ the set of original images used for training MPN. We also denote by $\Theta$ the set of model architectures used for generating attack data in $\mathcal{D}_{\mathrm{tr}}$. For ease of presentation, we express the training set of MPN as $\mathcal{D}_{\mathrm{tr}} = \{(\mathbf{z}, y)\}$ to omit the dependence on other factors.

Next, we elaborate on the construction of MPN (parameterized by $\phi$). We intend to make the architecture of MPN as simple as possible and make it different from the VM $\theta$. The rationale behind that has two folds. First, we would like to examine the *feasibility* of model parsing from adversarial attacks even forcing the *simplicity* of attribution network ($\phi$). Second, we would like to avoid the '*model attribute bias*' of $\phi$ when inferring VM attributes from adversarial attacks. Inspired by the above, we specify MPN by two simple networks: (1) multilayer perceptron (MLP) containing two hidden layers with 128 hidden units (0.41M parameters) (LeCun et al., 2015), and (2) a simple 4-layer CNN (ConvNet-4) with 64

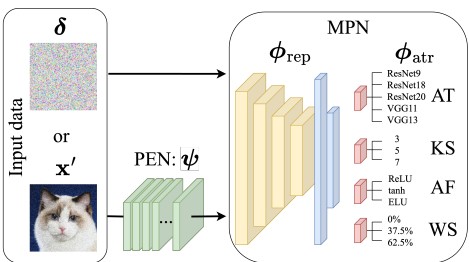

Figure 3: The schematic overview of our proposal. PEN (perturbation estimation network) is a pre-processing step to transform adversarial examples into perturbation-alike input data.

output channels for each layer, followed by one fully-connected layers with 128 hidden units and the attribution prediction head (0.15M parameters) (Vinyals et al., 2016). As will be evident later (**Fig. 4**), the model parsing accuracy of ConvNet-4 typically outperforms that of MLP. Thus, ConvNet-4 will be used as the MPN model by default. Given the datamodel setup, we next tackle the recognition problem of VM's attributes (AT, KS, AF, WS) via a multi-head multi-class classifier. We dissect MPN into two parts $\phi = [\phi_{\mathrm{rep}}, \phi_{\mathrm{atr}}]$, where $\phi_{\mathrm{rep}}$ is for data representation acquisition, and $\phi_{\mathrm{atr}}$ corresponds to the attribute-specific prediction head (*i.e.*, the last fully-connected layer in our design). Eventually, four prediction heads $\{\phi_{\mathrm{atr}}^{(i)}\}_{i=1}^4$ will share $\phi_{\mathrm{rep}}$ for model attribute recognition; see **Fig. 3** for a schematic overview of our proposal. The MPN training problem is then cast as

$$\underset{\phi_{\mathrm{rep}}, \{\phi_{\mathrm{atr}}^{(i)}\}_{i=1}^4}{\operatorname{minimize}} \quad \mathbb{E}_{(\mathbf{z},y)\in\mathcal{D}_{\mathrm{tr}}} \sum_{i=1}^{4} [\ell_{\mathrm{CE}}(h(\mathbf{z}; \phi_{\mathrm{rep}}, \phi_{\mathrm{atr}}^{(i)}), y_i)], \tag{1}$$

where $h(\mathbf{z}; \phi_{\mathrm{rep}}, \phi_{\mathrm{atr}}^{(i)})$ denotes the MPN prediction at input example $\mathbf{z}$ using the predictive model consisting of $\phi_{\mathrm{rep}}$ and $\phi_{\mathrm{atr}}^{(i)}$ for the $i$th attribute classification, $y_i$ is the ground-truth label of the $i$th attribute associated with the input data $\mathbf{z}$, and $\ell_{\mathrm{CE}}$ is the cross-entropy (CE) loss characterizing the error between the prediction and the true label.

**Evaluation methods.** Similar to training, we denote by $\mathcal{D}_{\mathrm{test}} = \{(\mathbf{z}(\mathcal{A}, \mathbf{x}, \theta), y(\theta)) \mid \mathbf{x} \in \mathcal{I}_{\mathrm{test}}, \theta \in \Theta\}$ the test attack set for evaluating the performance of MPN. Here the set of benign images $\mathcal{I}_{\mathrm{test}}$ is different from $\mathcal{I}_{\mathrm{tr}}$, thus adversarial attacks in $\mathcal{D}_{\mathrm{test}}$ are new to $\mathcal{D}_{\mathrm{tr}}$. To mimic the standard evaluation pipeline of supervised learning, we propose the following evaluation metrics.

*(1) In-distribution generalization*: The MPN testing dataset $\mathcal{D}_{\mathrm{test}}$ follows the attack methods ($\mathcal{A}$) and the VM specifications ($\Theta$) *same as* $\mathcal{D}_{\mathrm{tr}}$ but corresponding to different original benign images (*i.e.*, $\mathcal{I}_{\mathrm{test}} \neq \mathcal{I}_{\mathrm{tr}}$). The purpose of such an in-distribution evaluation is to examine if the trained MPN can infer model attributes encoded in new attack data given already-seen attack methods.

*(2) Out-of-distribution (OOD) generalization*: In addition to new test-time images, there could exist *attack/model distribution shifts* in $\mathcal{D}_{\mathrm{test}}$ due to using *new attack methods or model architectures*, leading to *unseen* attack methods ($\mathcal{A}$) and victim models ($\Theta$) different from the settings in $\mathcal{D}_{\mathrm{tr}}$.

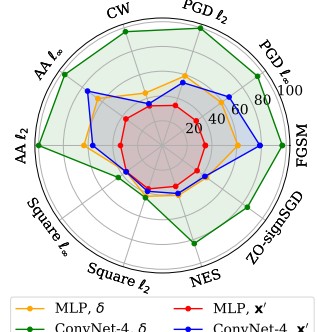

Figure 4: The in-distribution generalization of MPN using different formats of input data (adversarial perturbations $\delta$ vs. adversarial examples $\mathbf{x}'$) and parsing networks (ConvNet-4 vs. MLP). The generalization performance is measured by the averaged testing accuracy of attribute-specific classifiers; see Sec. 5. The attack data are generated with $\ell_\infty$ attack strength $\epsilon = 8/255$ and $\ell_2$ attack strength $\epsilon = 0.5$ on CIFAR-10. The VM architecture is fixed to ResNet-9.

In the rest of the paper, both in-distribution and OOD generalization capabilities will be assessed. Unless specified otherwise, the generalization of MPN stands for the *in-distribution generalization*.

**Perturbations or adversarial examples? The input data format matters for MPN.** Recall from Sec. 3 that an adversarial example, given by the linear model $\mathbf{x}' = \mathbf{x} + \delta$, relates to $\theta$ through $\delta$. Thus, it could be better for MPN to adopt *adversarial perturbations* ($\delta$) as the attack data feature ($\mathbf{z}$), rather than the indirect adversarial example $\mathbf{x}'$. **Fig. 4** empirically justifies our hypothesis by comparing

the generalization of MPN trained on adversarial perturbations with that on adversarial examples under two model specifications of MPN, `MLP` and `ConvNet-4`. We present the performance of MPN trained and tested on different attack types. As we can see, the use of adversarial perturbations ($\boldsymbol{\delta}$) consistently improves the test-time classification accuracy of VM attributes, compared to the use of adversarial examples ($\mathbf{x}'$). In addition, `ConvNet-4` outperforms `MLP` with a substantial margin.

Although Fig. 4 shows the promise of the generalization ability of MPN when trained and tested on adversarial perturbations, it may raise another practical question of how to obtain adversarial perturbations from adversarial examples if the latter is the only attack source accessible to MPN. To overcome this difficulty, we propose a *perturbation estimator network* (**PEN**) that can be jointly learned with MPN. Once PEN is prepended to the MPN model, the resulting end-to-end pipeline can achieve model parsing using adversarial examples as inputs (see Fig. 3). We use a denoising network, DnCNN (Zhang et al., 2017), to model PEN with parameters $\boldsymbol{\psi}$. PEN obtains perturbation estimates by minimizing the denoising objective function using the true adversarial perturbations as supervision. Extended from (1), we then have

$$\underset{\boldsymbol{\psi}, \boldsymbol{\phi}_{\mathrm{rep}}, \{\boldsymbol{\phi}_{\mathrm{atr}}^{(i)}\}_{i=1}^{4}}{\text{minimize}} \beta \mathbb{E}_{(\mathbf{x},\mathbf{x}')\in\mathcal{D}_{\mathrm{tr}}}[\ell_{\mathrm{MAE}}(g_{\boldsymbol{\psi}}(\mathbf{x}'), \mathbf{x}'-\mathbf{x})] + \mathbb{E}_{(\mathbf{x}',y)\in\mathcal{D}_{\mathrm{tr}}} \sum_{i=1}^{4}[\ell_{\mathrm{CE}}(h(g_{\boldsymbol{\psi}}(\mathbf{x}'); \boldsymbol{\phi}_{\mathrm{rep}}, \boldsymbol{\phi}_{\mathrm{atr}}^{(i)}), y_i)] \quad (2)$$

where $g_{\boldsymbol{\psi}}(\mathbf{x}')$ is the output of PEN given $\mathbf{x}'$ as the input, $\ell_{\mathrm{MAE}}$ is the mean-absolute-error (MAE) loss characterizing the perturbation estimation error, and $\beta > 0$ is a regularization parameter. Compared with (1), MPN takes the perturbation estimate $g_{\boldsymbol{\psi}}(\mathbf{x}')$ for VM attribute classification.

## 5 EXPERIMENTS

**Dataset curation.** We use standard image classification datasets (`CIFAR-10`, `CIFAR-100`, and `Tiny-ImageNet`) to acquire victim models, from which attacks are generated. We refer readers to Appendix A for details on victim model training and evaluation, as well as different attack setups. These VM instances are then leveraged to create MPN datasets (as described in Sec. 4). The attack types and victim model configurations have been summarized in Table 1 and 2. Thus, we collect a dataset consisting of adversarial attacks across 7 attack types generated from 135 VMs (configured by 5 architecture types, 3 kernel size setups, 3 activation function types, and 3 weight sparsity levels).

**MPN training and evaluation.** To solve problem (1), we train the MPN model using the SGD (stochastic gradient descent) optimizer with cosine annealing learning rate schedule and an initial learning rate of 0.1. The training epoch number and the batch sizes are given by 100 and 256, respectively. To solve problem (2), we first train MPN according to (1), and then fine-tune a pre-trained DnCNN model (Gong et al., 2022) (taking only the denoising objective into consideration) for 20 epochs. Starting from these initial models, we jointly optimize MPN and PEN by minimizing problem (2) with $\beta = 1$ over 50 epochs. In evaluation, we consider both in-distribution and OOD generalization. The generalization is measured by testing accuracy averaged over attribute-wise predictions, namely, $\sum_i (N_i \mathrm{TA}(i))/\sum_i N_i$, where $N_i$ is the number of classes of the model attribute $i$, and $\mathrm{TA}(i)$ is the accuracy of the classifier associated with the attribute $i$ (Fig. 3).

**In-distribution generalization of MPN is achievable. Table 3** shows the in-distribution performance of MPN trained using different input data formats (*i.e.*, adversarial examples $\mathbf{x}'$, PEN-estimated adversarial perturbations $\boldsymbol{\delta}_{\mathrm{PEN}}$, and true adversarial perturbations $\boldsymbol{\delta}$) given each attack type in Table 1. Here the choice of `AT`

Table 3: The in-distribution testing accuracy (%) of MPN trained using different input data formats (adversarial examples $\mathbf{x}'$, PEN-estimated adversarial perturbations $\boldsymbol{\delta}_{\mathrm{PEN}}$, and true adversarial perturbations $\boldsymbol{\delta}$) across different attack types on `CIFAR-10`, with $\ell_\infty$ attack strength $\epsilon = 8/255$, $\ell_2$ attack strength $\epsilon = 0.5$, and `CW` attack strength $c = 1$.

| Attack type / Input data | FGSM | PGD $\ell_\infty$ | PGD $\ell_2$ | CW | AA $\ell_\infty$ | AA $\ell_2$ | Square $\ell_\infty$ | Square $\ell_2$ | NES | ZO-signSGD |
|---|---|---|---|---|---|---|---|---|---|---|
| $\mathbf{x}'$ | 78.80 | 66.62 | 53.42 | 35.42 | 74.78 | 56.26 | 38.92 | 36.21 | 40.80 | 42.48 |
| $\boldsymbol{\delta}_{\mathrm{PEN}}$ | 94.15 | 83.20 | 82.58 | 64.46 | 91.09 | 86.89 | 44.14 | 42.30 | 58.85 | 61.20 |
| $\boldsymbol{\delta}$ | 96.89 | 95.07 | 99.64 | 96.66 | 97.48 | 99.95 | 44.37 | 44.05 | 83.33 | 84.87 |

(architecture type) is fixed to ResNet9, but adversarial attacks on `CIFAR-10` are generated from VMs configured by different values of `KS`, `AF`, and `WS` (see Table 2). As we can see, the generalization of MPN varies against the attack type even if model parsing is conducted from the ideal adversarial perturbations ($\boldsymbol{\delta}$). We also note that model parsing from white-box adversarial attacks (*i.e.*, `FGSM`, `PGD`, and `AA`) is easier than that from black-box attacks (*i.e.*, `ZO-signSGD`, `NES`, and `Square`). For example, the worst-case performance of MPN is achieved when training/testing on `Square` attacks. This is not surprising, since `Square` is based on random search and has the least dependence on VM attributes. In addition, we find that MPN using estimated perturbations ($\boldsymbol{\delta}_{\mathrm{PEN}}$) substantially outperforms the one trained on adversarial examples ($\mathbf{x}'$).

Table 4: In-distribution generalization (testing accuracy, %) of MPN given different choices of VMs and datasets, attack types/strengths, and MPN input data formats ($\mathbf{x}'$, $\boldsymbol{\delta}_{\mathrm{PEN}}$, and $\boldsymbol{\delta}$).

| Attack type | Attack strength | CIFAR-10 ResNet9 | | | CIFAR-10 ResNet18 | | | CIFAR-10 ResNet20 | | | CIFAR-10 VGG11 | | | CIFAR-10 VGG13 | | | CIFAR-100 ResNet9 | | | Tiny-ImageNet ResNet18 | | |
|---|---|---|---|---|---|---|---|---|---|---|---|---|---|---|---|---|---|---|---|---|---|---|
| | | $\mathbf{x}'$ | $\boldsymbol{\delta}_{\mathrm{PEN}}$ | $\boldsymbol{\delta}$ | $\mathbf{x}'$ | $\boldsymbol{\delta}_{\mathrm{PEN}}$ | $\boldsymbol{\delta}$ | $\mathbf{x}'$ | $\boldsymbol{\delta}_{\mathrm{PEN}}$ | $\boldsymbol{\delta}$ | $\mathbf{x}'$ | $\boldsymbol{\delta}_{\mathrm{PEN}}$ | $\boldsymbol{\delta}$ | $\mathbf{x}'$ | $\boldsymbol{\delta}_{\mathrm{PEN}}$ | $\boldsymbol{\delta}$ | $\mathbf{x}'$ | $\boldsymbol{\delta}_{\mathrm{PEN}}$ | $\boldsymbol{\delta}$ | $\mathbf{x}'$ | $\boldsymbol{\delta}_{\mathrm{PEN}}$ | $\boldsymbol{\delta}$ |
| FGSM | $\epsilon=4/255$ | 60.13 | 85.25 | 96.82 | 60.00 | 86.92 | 97.66 | 62.41 | 88.91 | 97.64 | 47.42 | 73.40 | 91.75 | 66.28 | 90.02 | 98.57 | 57.99 | 82.22 | 94.86 | 37.23 | 84.27 | 97.04 |
| | $\epsilon=8/255$ | 78.80 | 94.15 | 96.89 | 80.44 | 95.49 | 97.61 | 82.29 | 95.90 | 97.72 | 63.13 | 86.76 | 92.41 | 84.92 | 96.91 | 98.66 | 75.58 | 91.65 | 94.96 | 70.29 | 91.17 | 97.05 |
| | $\epsilon=12/255$ | 86.49 | 95.96 | 96.94 | 88.03 | 96.89 | 97.68 | 88.71 | 97.13 | 97.81 | 73.71 | 90.19 | 92.66 | 91.21 | 98.10 | 98.71 | 82.27 | 94.01 | 95.55 | 76.00 | 93.45 | 97.02 |
| | $\epsilon=16/255$ | 90.16 | 96.43 | 96.94 | 91.71 | 97.34 | 97.68 | 91.84 | 97.47 | 97.79 | 79.51 | 91.28 | 92.60 | 94.22 | 98.44 | 98.73 | 86.50 | 94.04 | 94.72 | 79.63 | 94.35 | 96.87 |
| PGD $\ell_{\infty}$ | $\epsilon=4/255$ | 50.54 | 76.43 | 96.02 | 56.94 | 79.45 | 96.96 | 55.01 | 80.05 | 97.49 | 39.33 | 66.38 | 91.84 | 57.12 | 81.18 | 98.29 | 42.27 | 72.62 | 92.65 | 35.48 | 76.56 | 97.18 |
| | $\epsilon=8/255$ | 66.62 | 83.20 | 95.07 | 73.29 | 87.29 | 95.38 | 67.49 | 86.19 | 96.18 | 56.62 | 81.14 | 92.78 | 69.16 | 88.46 | 97.22 | 59.71 | 79.55 | 90.43 | 61.85 | 82.90 | 96.05 |
| | $\epsilon=12/255$ | 76.65 | 89.73 | 94.91 | 81.73 | 91.67 | 95.55 | 76.41 | 90.16 | 95.67 | 70.56 | 88.92 | 94.13 | 78.67 | 92.93 | 97.26 | 70.86 | 85.31 | 91.28 | 73.82 | 88.80 | 96.38 |
| | $\epsilon=16/255$ | 75.58 | 86.95 | 91.28 | 82.46 | 90.19 | 93.19 | 76.58 | 87.79 | 92.50 | 72.13 | 87.23 | 91.85 | 78.28 | 90.20 | 94.66 | 71.29 | 82.35 | 86.84 | 73.19 | 85.02 | 93.54 |
| PGD $\ell_2$ | $\epsilon=0.25$ | 36.75 | 62.20 | 99.66 | 46.35 | 70.17 | 99.74 | 48.24 | 77.22 | 99.75 | 36.47 | 45.17 | 98.52 | 35.81 | 70.62 | 99.85 | 35.92 | 61.91 | 99.29 | 35.55 | 35.68 | 99.68 |
| | $\epsilon=0.5$ | 53.42 | 82.58 | 99.64 | 60.89 | 84.70 | 99.56 | 61.62 | 89.11 | 99.61 | 41.56 | 66.58 | 98.68 | 57.83 | 87.64 | 99.83 | 48.89 | 79.26 | 99.01 | 35.52 | 54.56 | 99.71 |
| | $\epsilon=0.75$ | 62.66 | 89.04 | 99.48 | 71.01 | 89.89 | 99.22 | 70.76 | 92.06 | 99.36 | 47.02 | 78.12 | 98.52 | 72.76 | 92.32 | 99.74 | 59.19 | 85.14 | 98.61 | 35.56 | 81.33 | 99.71 |
| | $\epsilon=1$ | 71.65 | 91.73 | 99.26 | 77.09 | 92.09 | 98.94 | 76.84 | 92.82 | 98.96 | 54.20 | 84.30 | 98.41 | 79.93 | 93.96 | 99.57 | 66.97 | 87.63 | 97.89 | 43.48 | 88.81 | 99.64 |
| CW | $c=0.1$ | 33.77 | 55.60 | 96.71 | 47.77 | 63.26 | 96.11 | 33.56 | 63.11 | 94.10 | 33.73 | 48.80 | 94.37 | 33.68 | 45.48 | 96.95 | 34.41 | 46.47 | 92.55 | 35.96 | 35.77 | 95.52 |
| | $c=1$ | 35.42 | 64.46 | 96.66 | 45.75 | 65.25 | 97.45 | 33.74 | 62.71 | 97.08 | 33.89 | 55.61 | 91.29 | 36.12 | 68.66 | 98.58 | 34.25 | 55.18 | 93.25 | 35.54 | 35.29 | 89.35 |
| | $c=10$ | 36.38 | 64.45 | 96.64 | 45.83 | 65.32 | 97.41 | 33.83 | 63.52 | 97.11 | 38.29 | 56.83 | 91.33 | 38.51 | 68.28 | 98.62 | 34.25 | 55.89 | 93.18 | 35.45 | 53.18 | 94.20 |

Extended from Table 3, **Fig. 5** shows the generalization of MPN when evaluated using attack data with different attack strengths. We observe that in-distribution generalization (corresponding to the same attack strength for the train-time and test-time attacks) is easier to achieve than OOD generalization (different attack strengths at test time and train time). A smaller gap between the train-time attack strength and the test-time strength leads to better generalization.

Table 3 and Fig. 5 focused on model parsing of adversarial attacks by fixing the VM architecture to ResNet9 on CIFAR-10, although different model attribute combinations lead to various ResNet9-type VM instantiations for attack generation. Furthermore, **Table 4** shows the in-distribution generalization of MPN under diverse setups of victim model architectures and datasets. The insights into model parsing are consistent with Table 3: (1) The use of true adversarial perturbations ($\boldsymbol{\delta}$) and PEN-estimated perturbations ($\boldsymbol{\delta}_{\mathrm{PEN}}$) can yield higher model parsing accuracy. And (2) inferring model attributes from white-box, gradient-based adversarial perturbations is easier, as supported by its over 90% testing accuracy. If adversarial examples ($\mathbf{x}'$) or estimated adversarial perturbations ($\boldsymbol{\delta}_{\mathrm{PEN}}$) are used for model parsing, the resulting accuracy gets better with a higher attack strength.

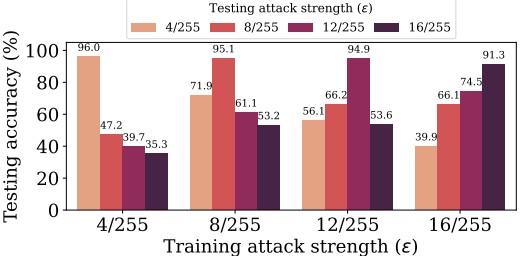

Figure 5: Accuracy (%) of MPN when trained on $\boldsymbol{\delta}$ generated by PGD $\ell_{\infty}$ using different attack strengths ($\epsilon$) and evaluated using different attack strengths as well. Other setups are consistent with in Table 3.

**OOD generalization of MPN becomes difficult vs. unseen, dissimilar attack types at testing time.** In **Fig. 6**, we present the generalization matrix of MPN when trained under one attack type (*e.g.*, PGD $\ell_{\infty}$ attack at row 1) but tested under another attack type (*e.g.*, FGSM attack at column 2) when adversarial perturbations are generated from the same set of ResNet9-based VMs (with different configurations of other model attributes) on CIFAR-10. The diagonal entries of the matrix correspond to the in-distribution generalization of MPN given the attack type, while the off-diagonal entries characterize OOD generalization when the test-time attack type is different from the train-time one.

**First**, we find that MPN generalizes better across attack types when they share similarities, leading to the following *generalization communities*: $\ell_{\infty}$ attacks (PGD $\ell_{\infty}$, FGSM, and AA $\ell_{\infty}$), $\ell_2$ attacks (CW, PGD $\ell_2$, or AA $\ell_2$), and ZOO-based black-box attacks (NES and ZO-signSGD). **Second**, Square attacks are difficult to learn and generalize, as evidenced by the low test accuracies in the last two

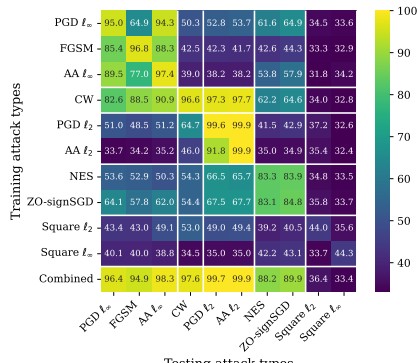

Figure 6: Performance (%) of MPN when trained on a row-specific attack but evaluated on a column-specific attack. The attack data are given by adversarial perturbations following the setup of Table 3 on ResNet9 and CIFAR-10. The 'combined' denotes the collection of: PGD $\ell_{\infty}$, PGD $\ell_2$, CW, and ZO-signSGD.

rows and the last two columns. This is also consistent with Table 3. **Third**, given the existence of generalization communities, we then combine diverse attack types (including PGD $\ell_{\infty}$, PGD $\ell_2$, CW, and ZO-signSGD) into an augmented MPN training set and investigate if such a data augmentation can boost the OOD generalization of MPN. The results are summarized in the **'combined' row of**

**Fig. 6**. As we expect, the use of combined attack types indeed makes MPN generalize better across all attack types except for the random search-based `Square` attack. In Fig. A1 of Appendix B, we find the consistent OOD generalization of MPN when PEN-based perturbations are used in MPN.

We refer readers to Appendix C for more results for parsing results of different model architectures and `AT` prediction.

**MPN to uncover real VM attributes of transfer attacks.** As a use case of model parsing, we next investigate if MPN can correctly infer the source VM attributes from transfer attacks when applied to attacking a different model (see Fig. 2). Given the VM architecture ResNet9, we vary the values of model attributes `KS`, `AF`, and `WS` to produce 27 ResNet9-type VMs. **Fig. 7** shows the transfer attack success rate (ASR) matrix (Fig. 7a) and

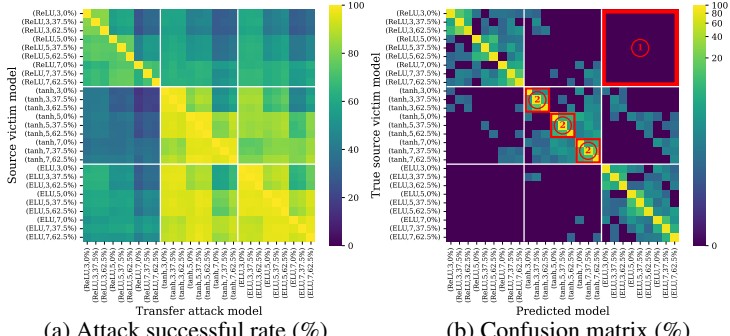

(a) Attack successful rate (%)      (b) Confusion matrix (%)

Figure 7: Model parsing of transfer attacks: Transfer attack success rate matrix (a) and model parsing confusion matrix (b). Given the architecture type ResNet9, the dataset `CIFAR-10`, and the attack type `PGD` $\ell_\infty$ (with strength $\epsilon = 8/255$), each model attribute combination (`AF`, `KS`, `WS`) defines a model instance to be attacked, transferred, or parsed.

the model parsing confusion matrix (Fig. 7b). Here the attack type is given by `PGD` $\ell_\infty$ attack with strength $\epsilon = 8/255$ on `CIFAR-10`, and the resulting adversarial perturbations (generated from different VMs) are used for MPN training and evaluation. In **Fig. 7a**, the off-diagonal entries represent ASRs of transfer attacks from row-wise VMs to attacking column-wise target models. As we can see, adversarial attacks generated from the ReLU-based VMs are typically more difficult to transfer to smooth activation function (ELU or tanh)-based target models. By contrast, given the values of `KS` and `AF`, attacks are easier to transfer across models with different weight sparsity levels.

**Fig. 7b** presents the confusion matrix of MPN trained on attack data generated from all 27 ResNet9-alike VMs. Each row of the confusion matrix represents the true VM used to generate the attack dataset, and each column corresponds to a predicted model attribute setting. The diagonal entries and the off-diagonal entries in Fig. 7b denote the correct model parsing accuracy and the misclassification rate on the incorrectly predicted model attribute configuration. As we can see, attacks generated from ReLU-based VMs result in a low misclassification rate of MPN on ELU or tanh-based predictions (see the marked region ①). Meanwhile, a high misclassification occurs for MPN when evaluated on attack data corresponding to different values of `WS` (see the marked region ②). The above results, together with our insights into ASRs of transfer attacks in Fig. 7a, suggest a connection between transfer attack and model parsing: *If attacks are difficult to transfer from the source model to the target model, then inferring the source model attributes from these attacks turns to be easy.*

**Model parsing vs. model robustness.** Further, we examine the adversarial robustness of victim models in the generalization of MPN; see **Fig. A3** in Appendix D. We find that adversarial attacks against robust VMs is harder to parse than attacks against standard VMs.

## 6 CONCLUSION

We studied the model parsing problem of adversarial attacks to infer victim model attributes, and approached this problem as a supervised learning task by training a model parsing network (MPN). We studied both in-distribution and out-of-distribution generalization of MPN considering unseen attack types and model architectures. We demystified several key factors, such as input data formats, backbone network choices, and VM characteristics (like attack transferability and robustness), which can influence the effectiveness of model parsing. Extensive experiments are provided to demonstrate when, how, and why victim model information can be inferred from adversarial attacks.

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

# Appendix

## A  VICTIM MODEL TRAINING, EVALUATION, AND ATTACK SETUPS

When training all `CIFAR-10`, `CIFAR-100`, and `Tiny-ImageNet` victim models (each of which is given by an attribute combination), we use the SGD optimizer with the cosine annealing learning rate schedule and an initial learning rate of 0.1. The weight decay is $5e-4$, and the batch size is 256. The number of training epochs is 75 for `CIFAR-10` and `CIFAR-100`, and 100 for `Tiny-ImageNet`. When the weight sparsity (`WS`) is promoted, we follow the one-shot magnitude pruning method Frankle and Carbin (2018); Ma et al. (2021) to obtain a sparse model. To obtain models with different activation functions (`AF`) and kernel sizes (`KS`), we modify the convolutional block design in the ResNet and VGG model family accordingly from 3, ReLU to others, *i.e.,* 5/7, tanh/ELU. Table A1 shows the testing accuracy (%) of victim models on different datasets, given any studied (`AF`, `KS`, `WS`) tuple included in Table 2. It is worth noting that we accelerate victim model training by using FFCV Leclerc et al. (2023) when loading the dataset.

Table A1: Victim model performance (testing accuracy, %) given different choices of datasets and model architectures.

| Dataset | AT | ReLU | | | | | | | | | tanh | | | | | | | | | ELU | | | | | | | | |
|---|---|---|---|---|---|---|---|---|---|---|---|---|---|---|---|---|---|---|---|---|---|---|---|---|---|---|---|---|
| | | 3 | | | 5 | | | 7 | | | 3 | | | 5 | | | 7 | | | 3 | | | 5 | | | 7 | | |
| | | 0% | 37.5% | 62.5% | 0% | 37.5% | 62.5% | 0% | 37.5% | 62.5% | 0% | 37.5% | 62.5% | 0% | 37.5% | 62.5% | 0% | 37.5% | 62.5% | 0% | 37.5% | 62.5% | 0% | 37.5% | 62.5% | 0% | 37.5% | 62.5% |
| CIFAR-10 | ResNet9 | 94.4 | 93.9 | 94.2 | 93.3 | 93.5 | 93.5 | 92.4 | 92.8 | 92.8 | 89.0 | 88.8 | 89.9 | 88.4 | 88.6 | 88.2 | 87.0 | 87.2 | 88.0 | 91.0 | 91.2 | 90.7 | 90.3 | 90.2 | 90.5 | 89.3 | 90.0 | 89.6 |
| | ResNet18 | 94.7 | 94.9 | 95.0 | 94.2 | 94.5 | 94.5 | 93.9 | 93.5 | 93.6 | 87.1 | 87.5 | 88.2 | 84.2 | 84.9 | 85.5 | 81.3 | 81.2 | 85.1 | 90.6 | 90.8 | 90.6 | 90.1 | 91.1 | 90.5 | 85.7 | 83.4 | 84.3 |
| | ResNet20 | 92.1 | 92.5 | 92.3 | 92.0 | 92.2 | 92.0 | 90.9 | 91.8 | 91.5 | 89.7 | 89.7 | 89.7 | 89.5 | 89.4 | 89.6 | 88.3 | 88.2 | 88.9 | 90.7 | 91.2 | 90.9 | 90.3 | 90.5 | 90.6 | 89.2 | 89.7 | 89.7 |
| | VGG11 | 91.0 | 91.1 | 90.4 | 89.8 | 89.9 | 89.4 | 88.2 | 88.4 | 88.0 | 88.7 | 89.1 | 88.9 | 87.2 | 87.6 | 87.6 | 87.0 | 86.8 | 87.0 | 89.4 | 89.5 | 89.5 | 89.0 | 88.2 | 88.5 | 88.6 | 87.1 | 87.2 |
| | VGG13 | 93.1 | 93.3 | 93.0 | 92.0 | 92.2 | 92.6 | 91.2 | 91.1 | 91.0 | 90.1 | 90.1 | 90.1 | 89.3 | 89.1 | 89.3 | 88.2 | 88.8 | 88.8 | 90.8 | 90.9 | 90.8 | 89.2 | 89.5 | 89.4 | 88.4 | 88.7 | 88.9 |
| CIFAR-100 | ResNet9 | 73.3 | 73.6 | 73.5 | 71.8 | 71.9 | 71.2 | 69.1 | 69.8 | 69.2 | 58.6 | 60.1 | 60.1 | 61.2 | 62.0 | 62.9 | 58.2 | 59.8 | 60.3 | 70.8 | 70.7 | 70.8 | 69.5 | 69.6 | 69.8 | 67.3 | 68.3 | 68.7 |
| | ResNet18 | 74.4 | 75.0 | 75.6 | 73.6 | 73.0 | 74.6 | 71.2 | 71.0 | 70.9 | 62.0 | 62.0 | 62.9 | 57.3 | 59.3 | 60.1 | 51.3 | 53.4 | 57.1 | 70.1 | 70.8 | 71.1 | 66.8 | 69.7 | 69.7 | 63.1 | 61.8 | 65.7 |
| | ResNet20 | 68.3 | 68.4 | 67.5 | 67.8 | 67.5 | 67.7 | 66.8 | 66.7 | 67.6 | 59.9 | 61.3 | 59.6 | 61.9 | 62.0 | 62.1 | 59.9 | 61.2 | 61.2 | 66.4 | 67.6 | 67.7 | 67.0 | 67.3 | 67.2 | 66.2 | 66.9 | 66.8 |
| | VGG11 | 68.3 | 68.4 | 67.7 | 65.2 | 65.7 | 65.8 | 62.4 | 62.0 | 62.6 | 65.2 | 65.5 | 65.5 | 63.6 | 63.6 | 63.9 | 62.1 | 61.8 | 62.5 | 66.2 | 66.5 | 65.9 | 64.6 | 64.0 | 64.6 | 61.5 | 62.3 | 61.9 |
| | VGG13 | 71.0 | 70.6 | 71.1 | 69.9 | 70.5 | 70.3 | 66.5 | 66.5 | 67.2 | 66.7 | 67.5 | 67.5 | 65.2 | 65.5 | 67.1 | 63.9 | 63.4 | 65.0 | 68.9 | 69.3 | 69.5 | 66.3 | 66.7 | 67.1 | 64.2 | 64.5 | 64.7 |
| Tiny-ImageNet | ResNet18 | 63.7 | 64.1 | 63.5 | 61.5 | 62.7 | 62.6 | 59.6 | 61.0 | 61.7 | 47.0 | 48.1 | 50.0 | 46.6 | 47.9 | 48.3 | 41.0 | 43.5 | 44.6 | 57.2 | 57.9 | 58.1 | 52.7 | 53.8 | 53.6 | 52.3 | 51.5 | 52.3 |

For different attack types, we list all the attack configurations below:

✦ `FGSM`. We set the attack strength $\epsilon$ equal to $4/255$, $8/255$, $12/255$, and $16/255$, respectively.

✦ `PGD` $\ell_\infty$. We set the attack step number equal to 10, and the attack strength-learning rate combinations as ($\epsilon = 4/255$, $\alpha = 0.5/255$), ($\epsilon = 8/255$, $\alpha = 1/255$), ($\epsilon = 12/255$, $\alpha = 2/255$), and ($\epsilon = 16/255$, $\alpha = 2/255$).

✦ `PGD` $\ell_2$. We set the step number equal to 10, and the attack strength-learning rate combinations as ($\epsilon = 0.25$, $\alpha = 0.05$), ($\epsilon = 0.5$, $\alpha = 0.1$), ($\epsilon = 0.75$, $\alpha = 0.15$), and ($\epsilon = 1.0$, $\alpha = 0.2$).

✦ `CW`. We use $\ell_2$ version CW attack with the attack conference parameter $\kappa$ equal to 0. We also set the learning rate equal to 0.01 and the maximum iteration number equal to 50 to search for successful attacks.

✦ `AutoAttack` $\ell_\infty$. We use the standard version of `AutoAttack` with the $\ell_\infty$ norm and $\epsilon$ equal to $4/255$, $8/255$, $12/255$, and $16/255$, respectively.

✦ `AutoAttack` $\ell_2$. We use the standard version of `AutoAttack` with the $\ell_2$ norm and $\epsilon$ equal to 0.25, 0.5, 0.75, and 1.0, respectively.

✦ `SquareAttack` $\ell_\infty$. We set the maximum query number equal to 5000 with $\ell_\infty$ norm $\epsilon$ equal to $4/255$, $8/255$, $12/255$, and $16/255$, respectively.

✦ `SquareAttack` $\ell_2$. We set the maximum query number equal to 5000 with $\ell_\infty$ norm $\epsilon$ equal to 0.25, 0.5, 0.75, and 1.0, respectively.

✦ `NES`. We set the query number for each gradient estimate equal to 10, together with $\mu = 0.01$ (*i.e.,* the value of the smoothing parameter to obtain the finite difference of function evaluations). We also set the learning rate by 0.0005, and the maximum iteration number by 500 for each adversarial example generation.

✦ `ZO-signSGD`. We set the query number for each gradient estimate equal to 10 with $\mu = 0.01$. We also set the learning rate equal to 0.0005, and the maximum iteration number equal to 500 for each adversarial example generation. The only difference between `ZO-signSGD` and `NES` is the gradient estimation method in ZOO. `ZO-signSGD` uses the sign of forward difference-based estimator while `NES` uses the central difference-based estimator.

## B OOD GENERALIZATION PERFORMANCE OF MPN ACROSS ATTACK TYPES WHEN PEN IS USED

Similar to Fig. 7, Fig. A1 shows the generalization performance of MPN when trained on a row-specific attack type but evaluated on a column-specific attack type when $\delta_{\text{PEN}}$ is given as input. When MPN is trained on the collection of four attack types PGD $\ell_\infty$, PGD $\ell_2$, CW, and ZO-signSGD (*i.e.*, the 'Combined' row), such a data augmentation can boost the OOD generalization except for the random search-based Square attack.

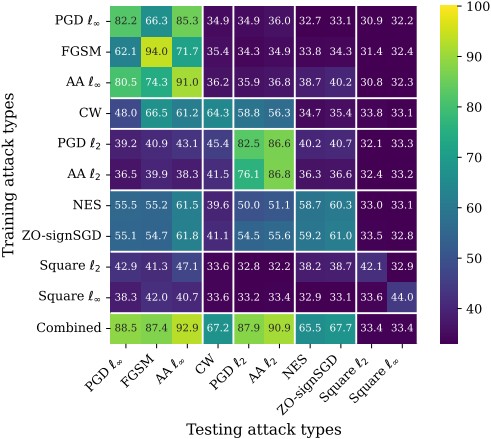

Figure A1: Generalization performance matrix of MPN when trained on a row-specific attack type but evaluated on a column-specific attack type given $\delta_{\text{PEN}}$ as input. The attack data are given by adversarial perturbations with strength $\epsilon = 8/255$ for $\ell_\infty$ attacks, $\epsilon = 0.5$ for $\ell_2$ attacks, and $c = 1$ for CW attack. The victim model architecture and the dataset are set as ResNet9 and CIFAR-10. The 'combined' row represents MPN training on the collection of four attack types: PGD $\ell_\infty$, PGD $\ell_2$, CW, and ZO-signSGD.

## C   MPN FOR ARCHITECTURE TYPE (AT) PREDICTION

**Fig. A2** demonstrates the generalization matrix of MPN when trained and evaluated using adversarial perturbations generated from different VM architectures (*i.e.*, different values of AT in Table 1) by fixing the configurations of other attributes (KS, AF, and WS). We observe that given an attack type, the in-distribution MPN generalization remains well across VM architectures. Yet, the OOD generalization of MPN (corresponding to the off-diagonal entries of the

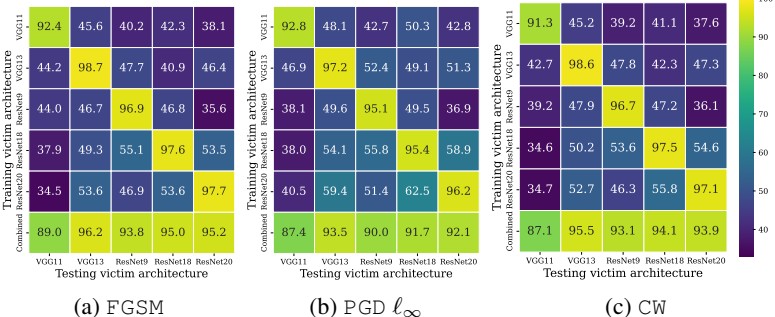

|                | (a) FGSM | (b) PGD $\ell_\infty$ | (c) CW |

Figure A2: Generalization matrix (%) of MPN when trained on attack data generated from a row-specific architecture but evaluated on attack data generated from a column-specific architecture. Both the train-time and test-time architectures share the same VM attributes in KS, AF, and WS. The attack type is specified by FGSM, PGD $\ell_\infty$, or CW on CIFAR-10, with the attack strength $\epsilon = 8/255$ for $\ell_\infty$ attacks and $c = 1$ for CW.

generalization matrix) rapidly degrades if the test-time VM architecture is different from the train-time one. Thus, if MPN is trained on data with AT classes as supervision, then the in-distribution generalization on AT retains.

MPN is also trained on (AT, AF, KS, WS) tuple by merging AT into the attribute classification task. We conduct experiments considering different architectures mentioned in Table 2 on CIFAR-10 and CIFAR-100, with $\delta$ and $\delta_{\text{PEN}}$ as MPN's inputs, respectively. We summarize the in-distribution generalization results in Table A2, Table A3, Table A4, and Table A5. Weighted accuracy refers to the testing accuracy defined in Sec. 5, *i.e.*, $\sum_i (N_i \text{TA}(i))/\sum_i N_i$, where $N_i$ is the number of classes of the model attribute $i$, and $\text{TA}(i)$ is the testing accuracy of the classifier associated with the attribute $i$ (Fig. 3). In the above tables, we also show the testing accuracy for each attribute, *i.e.*, $\text{TA}(i)$. Combined accuracy refers to the testing accuracy over all victim model attribute-combined classes, *i.e.*, 135 classes for 5 AT classes, 3 AF classes, 3 KS classes, and 3 WS classes. The insights into model parsing are summarized below: (1) MPN trained on $\delta$ and $\delta_{\text{PEN}}$ can effectively classify all the attributes AT, AF, KS, WS in terms of per-attribute classification accuracy, weighted testing accuracy, and combined accuracy. (2) Compared to AT, AF, and KS, WS is harder to parse.

Table A2: MPN performance (%) on different attack types given different evaluation metrics with adversarial perturbation $\delta$ as input on CIFAR-10.

| Metrics | Attack types | | | | | | | | | | | | | | |
|---|---|---|---|---|---|---|---|---|---|---|---|---|---|---|---|
| | FGSM | | | | PGD $\ell_\infty$ | | | | PGD $\ell_2$ | | | | CW | | |
| | $\epsilon=4/255$ | $\epsilon=8/255$ | $\epsilon=12/255$ | $\epsilon=16/255$ | $\epsilon=4/255$ | $\epsilon=8/255$ | $\epsilon=12/255$ | $\epsilon=16/255$ | $\epsilon=0.25$ | $\epsilon=0.5$ | $\epsilon=0.75$ | $\epsilon=1.0$ | $c=0.1$ | $c=1$ | $c=10$ |
| AT accuracy | 97.77 | 97.85 | 97.91 | 97.91 | 97.23 | 96.13 | 96.16 | 94.22 | 99.77 | 99.64 | 99.37 | 99.12 | 96.73 | 97.30 | 97.28 |
| AF accuracy | 95.67 | 95.73 | 95.79 | 95.71 | 95.86 | 95.26 | 95.77 | 94.05 | 99.51 | 99.36 | 99.04 | 98.68 | 95.12 | 94.84 | 94.68 |
| KS accuracy | 98.66 | 98.66 | 98.65 | 98.71 | 98.22 | 97.55 | 97.43 | 95.52 | 99.83 | 99.79 | 99.64 | 99.48 | 96.94 | 98.13 | 98.09 |
| WS accuracy | 87.16 | 87.16 | 87.29 | 87.52 | 84.36 | 79.99 | 80.01 | 71.68 | 98.51 | 97.83 | 96.86 | 95.57 | 88.42 | 85.28 | 85.03 |
| Weighted accuracy | 95.24 | 95.28 | 95.34 | 95.38 | 94.39 | 92.79 | 92.89 | 89.63 | 99.46 | 99.23 | 98.82 | 98.34 | 94.65 | 94.38 | 94.27 |
| Combined accuracy | 81.85 | 82.00 | 82.19 | 82.33 | 78.65 | 73.11 | 73.33 | 62.67 | 97.79 | 96.89 | 95.38 | 93.55 | 83.00 | 79.29 | 78.88 |

Table A3: MPN performance (%) on different attack types given different evaluation metrics with estimated perturbation $\delta_{\text{PEN}}$ as input on CIFAR-10.

| Metrics | Attack types | | | | | | | | | | | | | | |
|---|---|---|---|---|---|---|---|---|---|---|---|---|---|---|---|
| | FGSM | | | | PGD $\ell_\infty$ | | | | PGD $\ell_2$ | | | | CW | | |
| | $\epsilon=4/255$ | $\epsilon=8/255$ | $\epsilon=12/255$ | $\epsilon=16/255$ | $\epsilon=4/255$ | $\epsilon=8/255$ | $\epsilon=12/255$ | $\epsilon=16/255$ | $\epsilon=0.25$ | $\epsilon=0.5$ | $\epsilon=0.75$ | $\epsilon=1.0$ | $c=0.1$ | $c=1$ | $c=10$ |
| AT accuracy | 88.98 | 95.68 | 97.20 | 97.64 | 75.81 | 84.58 | 90.27 | 88.50 | 61.09 | 81.41 | 87.80 | 90.48 | 56.10 | 64.11 | 64.30 |
| AF accuracy | 83.48 | 92.21 | 94.56 | 95.22 | 74.95 | 85.04 | 90.72 | 89.81 | 57.62 | 76.90 | 83.95 | 87.36 | 54.61 | 58.77 | 58.98 |
| KS accuracy | 91.57 | 96.63 | 97.96 | 98.41 | 81.10 | 88.18 | 92.67 | 90.99 | 67.85 | 84.50 | 89.93 | 92.18 | 62.46 | 69.81 | 70.15 |
| WS accuracy | 69.99 | 81.42 | 84.92 | 86.59 | 56.07 | 63.92 | 70.19 | 64.80 | 50.09 | 67.26 | 74.02 | 77.70 | 46.40 | 47.53 | 47.77 |
| Weighted accuracy | 84.29 | 92.08 | 94.17 | 94.92 | 72.53 | 81.02 | 86.58 | 84.23 | 59.44 | 78.07 | 84.48 | 87.44 | 55.07 | 60.63 | 60.87 |
| Combined accuracy | 54.83 | 72.66 | 78.63 | 80.83 | 32.59 | 46.05 | 57.10 | 50.60 | 18.38 | 45.39 | 57.10 | 63.00 | 14.62 | 19.44 | 19.70 |

Table A4: MPN performance (%) on different attack types given different evaluation metrics with adversarial perturbation $\delta$ as input on CIFAR-100.

| Metrics | FGSM | | | | PGD $\ell_\infty$ | | | | PGD $\ell_2$ | | | | CW | | |
|---|---|---|---|---|---|---|---|---|---|---|---|---|---|---|---|
| | $\epsilon=4/255$ | $\epsilon=8/255$ | $\epsilon=12/255$ | $\epsilon=16/255$ | $\epsilon=4/255$ | $\epsilon=8/255$ | $\epsilon=12/255$ | $\epsilon=16/255$ | $\epsilon=0.25$ | $\epsilon=0.5$ | $\epsilon=0.75$ | $\epsilon=1.0$ | $c=0.1$ | $c=1$ | $c=10$ |
| AT accuracy | 97.70 | 97.76 | 97.76 | 97.75 | 97.03 | 95.40 | 95.23 | 92.52 | 99.59 | 99.29 | 98.91 | 98.50 | 93.84 | 96.23 | 96.30 |
| AF accuracy | 95.17 | 95.14 | 94.96 | 95.11 | 94.79 | 93.73 | 93.87 | 91.87 | 99.14 | 98.63 | 97.97 | 97.31 | 90.83 | 92.32 | 92.47 |
| KS accuracy | 97.66 | 97.65 | 97.69 | 97.62 | 96.75 | 95.16 | 94.44 | 91.25 | 99.62 | 99.43 | 99.16 | 98.70 | 93.11 | 95.77 | 95.81 |
| WS accuracy | 81.13 | 80.77 | 80.90 | 80.94 | 76.57 | 69.85 | 68.16 | 59.42 | 96.58 | 95.04 | 92.70 | 90.43 | 76.61 | 74.64 | 74.77 |
| Weighted accuracy | 93.60 | 93.54 | 93.53 | 93.55 | 92.11 | 89.52 | 88.97 | 85.02 | 98.85 | 98.27 | 97.43 | 96.56 | 89.34 | 90.67 | 90.76 |
| Combined accuracy | 75.08 | 74.76 | 74.82 | 74.95 | 69.72 | 61.27 | 59.31 | 48.37 | 95.27 | 93.06 | 89.89 | 86.73 | 67.19 | 66.24 | 66.56 |

Table A5: MPN performance (%) on different attack types given different evaluation metrics with estimated perturbation $\delta_{\text{PEN}}$ as input on CIFAR-100.

| Metrics | FGSM | | | | PGD $\ell_\infty$ | | | | PGD $\ell_2$ | | | | CW | | |
|---|---|---|---|---|---|---|---|---|---|---|---|---|---|---|---|
| | $\epsilon=4/255$ | $\epsilon=8/255$ | $\epsilon=12/255$ | $\epsilon=16/255$ | $\epsilon=4/255$ | $\epsilon=8/255$ | $\epsilon=12/255$ | $\epsilon=16/255$ | $\epsilon=0.25$ | $\epsilon=0.5$ | $\epsilon=0.75$ | $\epsilon=1.0$ | $c=0.1$ | $c=1$ | $c=10$ |
| AT accuracy | 88.17 | 95.25 | 96.92 | 97.45 | 72.40 | 82.48 | 88.11 | 85.47 | 62.77 | 80.01 | 85.88 | 88.33 | 47.31 | 51.80 | 52.48 |
| AF accuracy | 81.81 | 91.14 | 93.53 | 94.52 | 71.43 | 81.93 | 87.76 | 86.71 | 58.16 | 74.06 | 80.88 | 84.18 | 49.98 | 49.49 | 49.96 |
| KS accuracy | 88.62 | 94.92 | 96.58 | 97.12 | 76.97 | 84.74 | 88.78 | 86.46 | 69.38 | 84.09 | 88.68 | 90.56 | 56.07 | 59.68 | 59.72 |
| WS accuracy | 64.19 | 74.98 | 78.60 | 79.85 | 50.64 | 56.88 | 60.32 | 54.94 | 46.50 | 61.73 | 67.79 | 70.59 | 39.46 | 39.85 | 40.37 |
| Weighted accuracy | 81.76 | 89.95 | 92.19 | 92.98 | 68.51 | 77.36 | 82.22 | 79.40 | 59.71 | 75.69 | 81.53 | 84.12 | 48.08 | 50.43 | 50.90 |
| Combined accuracy | 47.75 | 65.27 | 71.05 | 73.27 | 25.56 | 37.49 | 45.28 | 38.97 | 16.27 | 38.87 | 49.04 | 54.06 | 7.31 | 9.20 | 9.59 |

## D   MODEL PARSING VS. MODEL ROBUSTNESS

We re-use the collected ResNet9-type victim models in Fig. 7, and obtain their adversarially robust versions by conducting adversarial training Madry et al. (2017) on CIFAR-10. Fig. A3 presents the generalization matrix of MPN when trained on a row-wise attack type but evaluated on a column-wise attack type. Yet, different from Fig. 6, the considered attack type is expanded by incorporating 'attack against robust model', besides 'attack against standard model'. It is worth noting that every attack type corresponds to attack data generated from victim models (VMs) instantiated by the combinations of model attributes KS, AF, and WS. Thus, the diagonal entries and the off-diagonal entries of the generalization matrix in Fig. A3 reflect the in-distribution parsing accuracy within an attack type and the OOD generalization across attack types. Here are two key observations. First, the in-distribution generalization of MPN from attacks against robust VMs is much poorer (see the marked region ①), compared to that from attacks against standard VMs. Second, the off-diagonal performance shows that MPN trained on attacks against standard VMs is harder to parse model attributes from attacks against robust VMs, and vice versa (see the marked region ②). Based on the above results, we posit that model parsing is easier for attacks generated from VMs with higher accuracy and lower robustness.

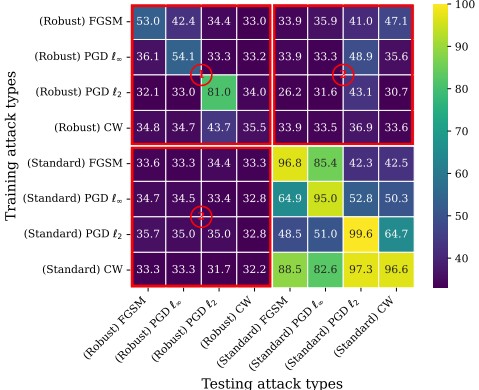

Figure A3:  Generalization performance (%) matrix of MPN across attack types, ranging from FGSM, PGD $\ell_\infty$, PGD $\ell_2$, and CW attacks against standard victim models to their variants against robust victim models, termed (Standard or Robust) Attack. Other setups are consistent with Fig. 6.

