# OpenReview forum: "Can Adversarial Examples Be Parsed to Reveal Victim Model Information?"
_ICLR.cc/2024/Conference — ICLR 2024 Conference Withdrawn Submission_

### Official Review · Reviewer_66Jb · 2023-10-22

**Soundness:** 2 fair
**Presentation:** 2 fair
**Contribution:** 2 fair
**Rating:** 3
**Confidence:** 4

**Summary:**

This work proposes a method to uncover information from a victim model through analysis of adversarial examples generated by it.
The authors use supervised training to recognize characteristics of a victim model (e.g., the activation function type).

**Strengths:**

+ interesting idea that might reveal some insights on transferability and on adversarial examples in general

**Weaknesses:**

- threat model is not clear
- the real challenge is to recognize unseen models
- no discussion on the limitations
- not tested on robust models
- poor visualization of figures and tables

**Questions:**

**Threat/defense model is not clear / unseen models / limitations.**
The initial part of the paper is not direct. The goal should be clear from the beginning. Is this an attack or a defense? Why would an attacker want to identify information of the victim model? Is it to identify an attacker and develop a detector (e.g., the cited RED approach)? The authors should clarify this aspect and even use the results achieved to demonstrate how this method should be used in practice (e.g., if it is used for a defense, they should then discuss how these results could be used).
The authors should specify how likely this is to happen and also what are the potential issues caused by this kind of attack.

Moreover, the authors assume that the victim model is in the training set, which is the greatest limitation of this work. Generating a dataset that contains all possible variations of model architectures and components would not scale. Thus, the authors should discuss this limitation more in detail and provide evidence that this limitation can be at least partially overcome with generalization abilities of this model.

This is an unlikely scenario. Additionally, the authors don't discuss limitations of the approach. For example, are there modifications to the models or to the attacks (e.g., randomness) that can prevent inferring these characteristics?

**Not tested on robust models / evaluation should be improved.**
The evaluation is performed with weak attacks and there are no insights on whether using robust models would prevent this attack. More details below:
* The number of steps for the PGD attacks is set to 10, which is a very low number for assuming convergence of the algorithm. The authors should perform the evaluation with more steps. The same applies to the other attacks.
* The authors should add adversarially-trained models to the experiments, to gather information on whether this technique would prevent the leakage of information from the adversarial examples generated with AT models
* the authors mix minimum-distnace attacks as CW and max-confidence attacks as PGD. The CW attack is used in a non conventional way, fixing the parameter c, which is in general search internally by the attack. While this might have been done to make the attack more similar to PGD (in fact, by removing the min-distance optimization), the authors should clarify this aspect in the experiments.

**Poor visualization.**
Almost all figures and tables in the paper are too small, framed in the text and poorly described. The authors should enhance both captions and descriptions and also enlarge the figures. More details are given below for individual issues.
* Figure 1 is difficult to understand, and the labels are not properly described. What are the axes of the radar plot? This figure should act as a graphical abstract, thus it's suggested to add more context on the method and outcomes.
* Figure 2 is even more unclear. The description is not sufficient and the figure is compressed in a very tiny space, making it even difficult to read. This figure should be enlarged and the description should be enhanced to better capture what this figure is depicting.
* all tables are too small and there is no highlighting of the most important results. While the highlighting is optional (even though it would improve clarity), the size of the tables should definitely be increased. The number inside the tables should have a font size at least comparable to the body of the manuscript.
* Figure 6 and 7 are too small and the labels are not easy to read. Moreover, the highlighted part of 7b should be described in the caption.



**Additional questions.**
- is there a reason why the authors test only the L-inf and L-2 perturbations? Would this approach generalize also to other types of attacks?

---

### Official Review · Reviewer_czv5 · 2023-10-25

**Soundness:** 3 good
**Presentation:** 3 good
**Contribution:** 2 fair
**Rating:** 5
**Confidence:** 4

**Summary:**

The paper provides a new perspective on the problem of adversarial examples (AE) against machine learning (ML) models. Specifically, the paper seeks to use AE to identify which ML model was used to create the AE themselves: in other words, the intention is determining which "victim model" (VM) an attacker used to craft the AE which will be later to launch an "adversarial attack". This new technique is defined as "parsing of adversarial examples", and it can be used in real security settings by defenders to uncover the strategies adopted by potential attackers that rely on AE to bypass their ML systems.

Hence, the paper first formalizes the problem of "parsing of AE", and then proposes a methodology to evaluate "the extent to which it is possible to parse the AE". This is done by crafting many AE (each by "attacking" a different ML model, and based on a different input sample), and then using the properties of such AE (i.e., attacked model and adversarial perturbation) to train a ML classifier---denoted as "model parsing network" (MPN). Then, the MPN is tested to infer the VM of "unseen" AE. Such testing is done both by considering "in-distribution AE" (i.e., drawn from the same classes of attacks seen during the training phase of the MPN) as well as on "out of distribution AE" (i.e., drawn from attacks that were not seen during training). Experiments on various ML models based on convolutional neural networks (CNN) trained on well-known image datasets (i.e., CIFAR and Tiny Image Net) validate the proposed approach.

**Strengths:**

+ Interesting research problem with practical relevance.
+ Extensive experiments
+ Well written, edited and presented


I particularly appreciated the clear use case described in Fig. 2. It is intriguing that nobody thought about this before: from a "defender" perspective, it is possible to use the findings of this paper to "steal" the model used "offline" by the attacker to craft the adversarial examples used "online" to attack the defender's model---provided, of course, that the defender can detect such adversarial examples. From this perspective, the proposed solution can be used as a form of "cyberthreat intelligence" or even for "cyber deception" -- and, potentially, even for "cyber forensics".

Overall, my sentiment is that the paper has high originality and novelty, high quality, and good clarity. The significance, however, is "mediocre", and the reasons will be outlined below.

**Weaknesses:**

## High-Level
- The "practical" usage requires to detect the adversarial examples first (which is far from trivial)
- Missing "intermediate" experiment on the detection of AE
- Only approachees the problem from a computer vision perspective (and only CNN)
- Some (fixable) terminology issues
- Unclear if the code will be disclosed
- Some parts are hard to interpret (and poor choice of OOD evaluation)

## Comment

Even though the paper has arguably many strengths (indeed, I was not able to point a clear "flaw" in the paper), I do have some concerns w.r.t. its significance and its overall contribution to the state-of-the-art. Indeed, while I do appreciate the novelty, I believe that the "practical value" of these findings to be low. In other words: how can the real-world benefit from this paper? On the one hand, I feel that the paper lacks a clear statement of how the proposed findings can be leveraged by future work; on the other hand, I feel that --to be usable-- these findings require some intermediate step (i.e., detecting AE) which is difficult to realise in practice.

Still, all such issues are fixable (to some extent) but some extra work is required. In what follows, I will provide detailed explanations and potential ways to address all the weaknesses outlined above.

### Practical usage

To my understanding, the only way to truly use the findings of this paper "in practice" is by detecting an adversarial example _first_, and then using it as basis to infer information on the corresponding VM. However, doing this is far from simple, as evidenced by many recent works (e.g., [F]). Hence, I wonder: what other usage is there for the method proposed in this paper? The text does mention that it opens new possibilities in the field of "robustness". To quote specific statements:

> If feasible, the insight into model parsing could offer us an in-depth understanding of the encountered threat model and inspire new designs of adversarial defenses and robust models.

> The insight into model parsing could offer us an in-depth understanding of the encountered threat model and inspire new designs of adversarial defenses and robust models.

...however, I question: how could this be done _in practice_? The paper does not elucidate how these insights can be leveraged for this purpose -- and, imho, I do not see any way to do so. For instance, from the viewpoint of a researcher (who is well-aware of the experimental pipeline, and hence "knows" what adversarial example is used to attack a given model), I do not see any way to leverage this. I invite the authors to discuss this---potentially by reducing the space given to the discussion of the results (which, personally, I found to be quite cumbersome to interpret).

### Missing "intermediate" experiment on the detection of AE ("How many samples are needed?")

Tieing back to the previous "open question", I am wondering how many samples are necessary "in practice" to understand that an attacker "truly" used a given VM. To my understanding, the process should be as follows: assume that an attacker, A, wants to evade a model M by means of some adversarial example x'. The defender is adopting a countermeasure, D, to detect adversarial examples. The attacker crafts the adversarial example x' by creating a surrogate of M (i.e., VM) and then transferring the adversarial examples x' to M. Let's call X' the set of all adversarial examples that bypassed VM and that are used to attack M.

In the experiments, the assumption is that X' is used to "test" (or "train") the proposed MPN. However, in reality, only a subset of X' would be detected by D (given that detection mechanisms are not perfect). Let's call S such a subset. The defender can then do the "forensic" analysis only on S (and not on X').

My question is: what is the performance of MPN on S? And, more importantly, how large is such S? Because if S only has, say, 10 samples (whereas X' has 1000) then it may be unpractical to derive any sound conclusion from the accuracy of an MPN tested on S.

My stance is that, from a practical viewpoint, it would be insightful to perform an additional analysis focused on determining the feasibility of the approach by assuming an "adversarial detection method" used to collect the samples that will then be further analysed. This would resemble a realistic scenario that allows to shed light on the attacker tactics. Perhaps a discussion of such a "practical use-case" could be added in an appendix (I would love to read something like this!)

### Only image data (and only Deep Learning)

The paper only considers ML applications within the deep learning (DL) paradigm, and -- specifically -- onl Convolutional Neural Networks that analysed Image data. However, as shown by a plethora of papers, the breadth of domains in which ML has found applications (and which can hence be attacked via adversarial examples) is much broader. Such a limited scope is, in my opinion, a limitation of this analysis: I would be delighted to see the proposed mechanism be applied in a different context, potentially entalining ML models that have nothing to do with neural networks and/or which analyse a different data type (e.g., time series; or tabular data).

### Some (fixable) terminology issues

The paper consistently relies on terminology that has been recently questioned [A].

For instance, I invite the authors to revise the usage of "black/white-box" and use "zero/perfect knowledge" instead (similarly, WB and BB should become PK and ZK).

Furthermore, the paper many times mentions the term ```adversarial attacks```. According to [A], this is a thautology that is confusing to many. I endorse the authors to revise any occurrence of such term.

Moreover, there are a series of statements that I invite the authors to revise. In what follows, I will directly quote such statements and express my concerns.

> have emerged as a primary security concern of ML in a wide range of vision applications

I disagree. Actually, there is evidence suggesting that adversarial ML attacks are not among the priorities of practitioners see [A,B,C,D]. At most, they are now seen as a means to achieve "robustness" rather than "security": these two goals, while similar, are orthogonal (even a very recent work mentioned "false state of robustness" [E])

> The adversarial attack (a.k.a, adversarial example)

I disagree with such wording. An adversarial example is "not" an attack. The "attack" is the process of crafting the adversarial example with the specific purpose of subverting an ML model. Potentially, an adversarial attack entails the creation of multiple adversarial examples. Hence I endorse the authors to revise this association, which is imprecise from a security perspective (see [A]).

> with full access to $\theta$

What does "full access" means? I endorse the authors to look into the "terminology issues" discussed in [A].

Finally, I must note that the caption of figures reports "Figure", but in the text it is referenced as "Fig."

### Some parts are hard to interpret (and poor choice of OOD evaluation)

This is just a personal comment, but I felt that the presentation of the results and of the experimental methodology to be convoluted. For instance, I still did not fully understand how the OOD experiments have been carried out: taking Figure A1 as an example, its caption reports

> performance of MPN when trained on a row-specific attack type, but evaluated on a column-specific attack type

Let's take the cell in the first row (PGD-l_inf) and second column (FGSM). Does it mean that the MPN is trained on PGD, and tested on FGSM? If so, does it mean that 66.3% of the times the MPN was able to infer the correct VM?

At the same time, the description in the main text is also vague:

> In addition to new test-time images, there could exist attack/model distribution shifts in $\mathcal{D}test$ due to using new attack methods or model architectures, leading to unseen attack methods ($\mathcal{A}$) and victim models ($\Theta$) different from the settings in $\mathcal{D}_{tr}$.

While I do see some reason in assuming such a OOD setting (because, in reality, a defender does not know the spectrum of attack techniques used to craft the AE), there is also another one which is _more interesting_ to consider: the case in which the OOD entails VM that are not known a-priori.

For instance, what would happen if the attacker crafted the AE using a CNN with a different architecture than the ones envisioned in Table 2? I feel that not investigating this case is a limitation from a practical perspective, given that the true goal of the "AE parsing" should be to infer the VM---but the space of potential VM is infinite, meaning that a 100% match is unfeasible to achieve in practice. Hence, the point is (IMHO!) assessing "how close to the real VM a given prediction is). For instance, assume that the attacker used ResNet18 but that ResNet18 is not included among the "classes" used to train the MPN: I would expect the MPN to predict "ResNet20" (since, among the remaining VM, it is the most similar); however, if the MPN predicts "VGG13", then the performance is "bad".

My stance is that the paper makes a "closed world assumption", which is unrealistic in practice. This, alongside the other weaknesses mentioned above, raises some skepticism on the overall "value" of this methodology (which, I stress, is novel and potentially interesting to some researchers!).

#### EXTERNAL REFERENCES

[A]: Apruzzese, Giovanni, et al. "“Real Attackers Don't Compute Gradients”: Bridging the Gap Between Adversarial ML Research and Practice." 2023 IEEE Conference on Secure and Trustworthy Machine Learning (SaTML). IEEE, 2023.

[B]: Boenisch, Franziska, et al. "“I Never Thought About Securing My Machine Learning Systems”: A Study of Security and Privacy Awareness of Machine Learning Practitioners." Proceedings of Mensch und Computer 2021. 2021. 520-546.

[C]: Kumar, Ram Shankar Siva, et al. "Adversarial machine learning-industry perspectives." 2020 IEEE Security and Privacy Workshops (SPW). IEEE, 2020.

[D]: Bieringer, Lukas, et al. "Industrial practitioners' mental models of adversarial machine learning." Eighteenth Symposium on Usable Privacy and Security (SOUPS 2022). 2022.

[E]: Pintor, Maura, et al. "Indicators of attack failure: Debugging and improving optimization of adversarial examples." Advances in Neural Information Processing Systems 35 (2022): 23063-23076.

[F]: Tramer, Florian. "Detecting adversarial examples is (nearly) as hard as classifying them." International Conference on Machine Learning. PMLR, 2022.

**Questions:**

I liked the paper, and I do see abundant scientific merit in the research discussed in this paper. However, after reading the paper I am left with more questions than before.

I hence endorse the authors to (i) read my review carefully, and point out any mistakes / inaccuracies that I may have made; then, (ii) I invite the authors to answer the following questions.


1) Was I correct in hypothesizing that the OOD evaluation assumes a "closed world scenario"? If so, would the authors be able to carry out some additional experiments in an "open world" setting, by removing some of the "classes" of VM from the training set and seeing how the resulting MPN fares in these conditions?

2) Was I correct in stating that, from a "defensive security" viewpoint, using the proposed AE parsing requires to detect the AE first? If so, would the authors be able to carry out some additional experiments on a set of AE  that have been "detected" by a state-of-the-art AE detection method?

3) Can the authors elaborate on potential ways in which the proposed findings can be used to improve the generic robustenss of ML models against AE?

4) Can the authors carry out experiments covering a different domain / ML model type?

5) Are the authors going to publicly release the codebase used for this paper? If no, why not?

Do note that, depending on the answers given, my score may increase substantially.

---

### Official Review · Reviewer_FJLK · 2023-10-30

**Soundness:** 3 good
**Presentation:** 3 good
**Contribution:** 3 good
**Rating:** 6
**Confidence:** 2

**Summary:**

This paper proposes a method called 'model parsing of adversarial attacks' that aims to reveal the characteristics of the victim model used to generate the attack. The authors show that their approach can successfully infer the model attributes from unseen attacks. The paper discusses the challenges in uncovering hidden information in adversarial attacks on machine learning models and presents the architecture and features of the model parsing network (MPN). The authors also demonstrate how the model parsing approach can be used to uncover the source victim model attributes from transfer attacks and show the connection between model parsing and attack transferability. Comprehensive experiments are carried out to support the authors' claims.

**Strengths:**

- The paper presents a novel approach to model parsing from the perspective of adversarial attacks.  This research direction, to the best of my knowledge, is rarely studied before.
- The authors provide a clear and detailed description of their methodology and experiments.
- The authors not only consider the in-distribution generalization but also the out-of-distributional scenarios where new attacks and models are used.
- The authors provide a comprehensive evaluation of their approach over a wide range of attacks and victim models.

**Weaknesses:**

- I'm unfamiliar to the research field of model parsing. I wonder if the authors can provide some comparisons to existing works that are closely related to model parsing?

**Questions:**

Please see weaknesses for details.

---

### Official Review · Reviewer_8ej7 · 2023-11-01

**Soundness:** 2 fair
**Presentation:** 4 excellent
**Contribution:** 2 fair
**Rating:** 5
**Confidence:** 4

**Summary:**

The paper proposes the task of estimating architecture choices for a neural model-- namely four dimensions (architecture type, kernel size, activation function, and weight sparsity)-- given an adversarial example that fools the model. For this purpose, the authors hypothesize that the adversarial perturbation, developed using various white-box and blackbox techniques, show correlation to the aforementioned model attributes being predicted and propose a neural network architecture to learn this mapping. The authors develop a training and test sets using existing attacks and model architectures and evaluate a trained model's performance on in-domain (attacks developed using attack methods and model architectures used at training time) and out-of-domain (OOD) performance. Particular neural networks (CNN-based) are able to perform well but often need only the perturbation vector $\delta$ as opposed to the entire adversarial image $x'$. Given the former constitutes a less realistic threat model, the authors train a 2nd neural model (jointly) to estimate the perturbation vector $\delta$. The experiments show (1) approximation works less effectively compared to using the perturbation vector direction, (2) OOD generalization is difficult and prediction showcases better generalization over similar attack types, and (3) a greater different between source and target models, implying less transferability, implies source model attacks leak more confident information about source model attributes.

**Strengths:**

1. The proposed problem is a novel one and combines (1) inferring of attack information given adversarial examples and (2) inferring attributes of a virtual model using non-perturbed data and model logits.
2. The paper is well-presented helping users clearly understand the contribution. The introducing classification network, addition of PEN objectives, and evaluation of in and out of domain settings is well-motivated and written in a logically fluent manner.

**Weaknesses:**

1. The question of whether a strong correlation actually exists such that one can effectively a classification function that can predict a victim model attributes given adversarial instances is perhaps the elephant-in-the-room. Showcasing that a developed neural network is able to predict with $x%$ accuracy the model parameters empirically and showing that the attack development optimization can involve the model parameter $\theta$ is a weak motivation.
2. Transferability of attacks is well-known in literature [1] and has been a major point in the design of defense mechanisms that use multiple victim models [2,3]. This sort of hints that the function form an attack to VM parameters is a one to many task. Yet, the classification task is designed as a one-to-one function.
3. One wonders what happens in the context of Universal Perturbations [4]? While these were originally designed to be a single perturbation vectors that can be used across the test set, what happens when you consider a generalized version that is effective across model architectures [5]?
4. In the black-box setting, what happen when the distilled model has a different architecture & VM attributes than the parent/target model? Precisely, will the learned attack vector showcase characteristics of the distilled model or the target model?
5. The results with the PEN network estimates $\delta_{PEN}$ seems quiet worse-off compared to using the perturbation $\delta$ (eg. 62.2 _vs._ 99.66 for PGD $l_2, \epsilon=0.25$ on ResNet9, 66.38 _vs._ 91.84 for $l_\infty, \epsilon=4/255$ on VGG 11, etc.) Is there a good analysis on why this is these cases? Is it a problem with the $PEN$ network? Why is the disparity so high for small perturbations like $\epsilon=4/255, 0.25$ (where I was expecting higher probability of unique signatures for model types) and much less for higher perturbations like $\epsilon=16/255, 1$?

> [1] Szegedy, C., Zaremba, W., Sutskever, I., Bruna, J., Erhan, D., Goodfellow, I., & Fergus, R. (2013). Intriguing properties of neural networks. arXiv preprint arXiv:1312.6199.

> [2] Sengupta, S., Chakraborti, T., & Kambhampati, S. (2019). Mtdeep: boosting the security of deep neural nets against adversarial attacks with moving target defense. In Decision and Game Theory for Security: 10th International Conference, GameSec 2019, Stockholm, Sweden, October 30–November 1, 2019, Proceedings 10 (pp. 479-491). Springer International Publishing.

> [3] Adam, G. A., Smirnov, P., Duvenaud, D., Haibe-Kains, B., & Goldenberg, A. (2018). Stochastic combinatorial ensembles for defending against adversarial examples. arXiv preprint arXiv:1808.06645.

> [4] Moosavi-Dezfooli, S. M., Fawzi, A., Fawzi, O., & Frossard, P. (2017). Universal adversarial perturbations. In Proceedings of the IEEE conference on computer vision and pattern recognition (pp. 1765-1773).

> [5] Chaubey, A., Agrawal, N., Barnwal, K., Guliani, K. K., & Mehta, P. (2020). Universal adversarial perturbations: A survey. arXiv preprint arXiv:2005.08087.

**Questions:**

See above.

---

### Official Review · Reviewer_9QLD · 2023-11-08

**Soundness:** 1 poor
**Presentation:** 2 fair
**Contribution:** 1 poor
**Rating:** 3
**Confidence:** 3

**Summary:**

The authors introduce techniques to infer details about a target model (architecture details, activation function choices, etc.) using signals from adversarial attacks. The authors observe high inference accuracy using a model parsing network. They also illustrate the utility of model parsing in identifying the source victim model of transfer attacks and establish a link between model parsing and attack transferability.

**Strengths:**

- Experimental design is thorough, with experiments over multiple attributes and datasets. Ablation studies over the right variables also help improve confidence in the proposed inference approach.

- The proposed technique has potential to be an enabler for stronger attacks in other fields. For instance, knowing the exact target architecture could be useful to construct stronger security (adversarial) and privacy-related (e.g. membership inference) attacks.

**Weaknesses:**

- The threat model does not make much sense to me. For instance, if the attacker uses an attack like PGD to construct adversarial examples, it has to have white-box access to the model anyway. Then what is the point of having inference for attributes of the model that are anyway known? Such inference would only make sense for black-box attacks.

- Table 1: The selection of attacks seems arbitrary. They are not state-of-the-art (there are dozens of attacks that are better, such as MIDIFGSM. See [this survey](https://arxiv.org/pdf/2310.17534.pdf) for an extensive analysis). Also, for these attacks and their hyper-parameters, please also mention the number of iterations the attacks are run for, etc.

- Section 4: "_We first create a model parsing dataset by collecting adversarial attack instances against victim models._" I am confused about the setup- you're using adversarial examples generated against victim models, to learn their attributes? Shouldn't the models at test time be different from the ones you used in training? Assuming these models are indeed trained (or, at least, available), a simple approach I can think of is to just evaluate all of them (on some data sample X) and pick the one that seems to give the best performance on target model. Why go through this hassle?

- Section 5: "_The generalization is measured by testing accuracy averaged over attribute-wise predictions, namely..._" Odd metric- is this standard? Why is the number of classes part of the metric? Also, should ideally report attribute-wise classification accuracies as well.

- Pg9: "_PN to uncover real VM attributes of transfer attacks._" - So the results before used the same model as the one used in training the attribute-inference model? No wonder it works so well - this experiment right here should be the main focus of analysis.

## Minor Issues

- Pg 3 (and other references): 'VM' is an unnecessarily confusing abbreviation. Please use 'victim model' or 'target model'
- Pg 3, RED: Seems to be a very new and niche concept, and readers should not be expected to know about it much- please explain how this is useful, what can a victim do if it knows adversary is running untargeted adv attack, etc.
- Pg 5, "_(Problem statement) Is it possible to infer victim model information from adversarial attacks? And what factors will influence such model parsing ability?_" I can reason about the motivation for this (knowledge could allow using closer local models, etc.), but should be stated more explicitly in the paper to motivate the threat model. One concrete way to do this would be: assume attribute inference works perfectly, how much of a jump in attack success can it get? Compare it with an attack that has the same query/compute budget, and see if this approach works better.
- Pg 6, "thus adversarial attacks in Dtest are new to Dtr" - How does that matter? It should be the models that are new.
- Table 4: Looking at results across makes me think the choice of adv attack will not matter much, but should include experiments nonetheless to confirm.
- Figure 7: Way too small text in heatmap- okay to only include representative ones (and leave bigger version for Appendix) but please make text labels bigger

**Questions:**

Please see 'Weaknesses' above